# Validity of an iPhone App to Detect Prefrailty and Sarcopenia Syndromes in Community-Dwelling Older Adults: The Protocol for a Diagnostic Accuracy Study

**DOI:** 10.3390/s22166010

**Published:** 2022-08-11

**Authors:** Alessio Montemurro, Juan D. Ruiz-Cárdenas, María del Mar Martínez-García, Juan J. Rodríguez-Juan

**Affiliations:** 1Physiotherapy Department, Faculty of Health Sciences, Universidad Católica de Murcia, Campus de los Jerónimos, 30107 Murcia, Spain; 2Cystic Fibrosis Association of Murcia, Av. de las Palmeras, 37, 30120 Murcia, Spain; 3Physiotherapy Department, Facultad de Medicina, Universidad de Murcia, Campus Espinardo, 30100 Murcia, Spain

**Keywords:** aging, sit-to-stand, frail, sarcopenia, functional capacity, muscle power, chair rise, smartphone

## Abstract

Prefrailty and sarcopenia in combination are more predictive of mortality than either condition alone. Early detection of these syndromes determines the prognosis of health-related adverse events since both conditions can be reversed through appropriate interventions. Nowadays, there is a lack of cheap, portable, rapid, and easy-to-use tools for detecting prefrailty and sarcopenia in combination. The aim of this study is to validate an iPhone App to detect prefrailty and sarcopenia syndromes in community-dwelling older adults. A diagnostic test accuracy study will include at least 400 participants aged 60 or over without cognitive impairment and physical disability recruited from elderly social centers of Murcia (Spain). Sit-to-stand muscle power measured through a slow-motion video analysis mobile application will be considered as the index test in combination with muscle mass (calf circumference or upper mid-arm circumference). Frailty syndrome (Fried’s Phenotype) and sarcopenia (EWGSOP2) will both be considered as reference standards. Sensibility, specificity, positive and negative predictive values and likelihood ratios will be calculated as well as the area under the curve of the receiver operating characteristic. This mobile application will add the benefit for screening large populations in short time periods within a field-based setting, where space and technology are often constrained (NCT05148351).

## 1. Introduction

Frailty and sarcopenia are syndromes related to the biological ageing process which can lead to adverse health-related consequences. Although not formally recognized as a disease, frailty is defined as a biological state characterized by reduced muscle strength, endurance, and physiological function that increases vulnerability to stressor events including loss of autonomy, disability, falls, hospitalizations, and deaths [1]. Although this syndrome seems to overlap with sarcopenia, in fact they are not synonyms. Indeed, sarcopenia is defined as a generalized and progressive loss of strength, skeletal muscle mass, and physical function [2] and it has been considered a precursor syndrome for developing frailty [3]. However, frailty is a multidimensional geriatric syndrome which not only includes physical but also cognitive and social dimensions. While sarcopenia is now formally recognized as a muscle disease with an ICD-10-MC Diagnosis Code, there is still a need for an ICD code for frailty to promote clinicians, public health agencies, and governments to report diagnosis, treatment strategies, and disease prevalence with enough accuracy [4].

Prevalence studies in European community-dwelling older adults have shown, with regional and criteria-related variations for diagnosis, that approximately 22% of them can be classified as sarcopenic [5], while 17% of them can be classified as frail, with Spain being among the countries with the highest prevalence rates [6]. However, few studies have analyzed the prevalence of both syndromes in combination and have reported prevalence rates of 2.8% and 3.6% in Australian [7] and Japanese [8] community-dwelling older adults, respectively, whereas the prevalence is even higher in the Spanish population, showing rates ranging from 8.2% to 15.7% depending on the diagnostic criteria used [9]. While both syndromes share many common features linked to the ageing process, recent studies have demonstrated that their coexistence increases the incidence of recurrent falls, poor health-related quality of life, and mortality compared to those neither frail nor sarcopenic [7,8]. Although sarcopenia and frailty are not the same condition, their association might respond to the existence of different clinical forms of frailty, i.e., those with or without sarcopenia, with different pathophysiological processes and increased risk of adverse outcomes [7,9,10]. Therefore, assessing sarcopenic status once a patient is diagnosed of frailty would have potential prognostic and diagnostic consequences [9].

Given the growing importance of the early identification of frailty (prefrailty status) and sarcopenia to prevent subsequent functional decline, disability, and the increased mortality risk, a clinical affordable easy-to-use diagnostic tool for detecting prefrailty and sarcopenia syndromes in combination might have implications for prognosis, optimizing care, and planning interventions for community-dwelling older adults. In this regard, time needed to stand from a chair is considered a biomarker of mortality and frailty in older people [1,11], as well as a reliable diagnostic criteria for probable sarcopenia [2]. Although time to complete the sit-to-stand test is the primary measure of function and can be easily measured with a stopwatch, muscle power derived from inertial measurement unit sensors has been associated with frailty and has a greater discriminatory capacity to detect frailty compared to the time to complete the test [12]. Similarly, kinematics data related to muscle power obtained from ground reaction forces have been associated with sarcopenia syndrome and showed an excellent discriminatory ability for detecting sarcopenia in a sample of 627 community-dwelling older adults [13]. However, although this technology seems to be sensitive to detect both syndromes, it is not readily available or affordable in clinical or field-based testing environments and some of the current methods remain complex, difficult and time consuming to analyze.

Recent advances in hand-held technology offer an opportunity to assess muscle power during sit-to-stand test through a smartphone mobile application as valid as a force plate and 3D motion capture cameras [14,15] with the advantage of acquiring muscle power with a short assessment duration and automatic data processing response. Indeed, the purpose of this study is to develop and validate an update of the *Sit-to-Stand* App for detecting prefrailty and sarcopenia syndromes in combination in community-dwelling older adults. Since several studies have shown that low muscle mass measured by calf circumference or mid upper-arm circumference has been associated with frailty and sarcopenia syndromes [16,17], one could expect that the incorporation of these two measures, muscle power and muscle mass, into a single diagnostic tool integrated in a mobile application will result in a portable and unique easy-to-use tool capable of early detecting both syndromes in combination with enough accuracy. This mobile application will add the benefit for screening large populations in short time periods within a field-based setting, where space and technology are often constrained.

## 2. Materials and Methods

### 2.1. Study Design and Protocol Registration

This is a protocol for a retrospective diagnostic accuracy study. Standards for Reporting of Diagnostic Accuracy Studies (STARD) [18] have been used to ensure the completeness and the transparency of this protocol (see Appendix A). This study has been registered on ClinicalTrial.gov (21/12/08): NCT0514835. This study was approved by the Ethical Committee of Catholic University of Murcia (CE022108) and is in accordance with the principles of the Declaration of Helsinki.

### 2.2. Eligibility Criteria

Participants aged 60 or over will be enrolled at elderly social centers of the city of Murcia (Region of Murcia, Spain). Participants self-reporting severe cardiovascular problems such as heart valve disease, uncontrolled heart rhythm problems, automatic defibrillator, and pacemakers will be excluded. Additionally, participants with a Barthel score less than 90, less than 3 points in the Mini-Cog test, or those who are unable to stand up from a chair without assistance will be excluded from this study. All participants will provide written informed consent to participate in this study.

### 2.3. Participants Recruitment

Centers will be contacted and informed for participation in the development of this study. After obtaining consent from the social centers, each potentially eligible subject will be contacted via telephone or face to face in order to individually provide a detailed explanation of the assessment procedure, as well as to obtain his/her written consent after receiving oral and written information about this study. Participants will be informed that they can withdraw at any time from this study. Participants’ recruitment will be carried out with a convenience model.

### 2.4. General Data Collection

All data collection will be performed at the same day. Data relative to age and sex of the subjects will be collected in form of self-reported questionnaire. Moreover, the following information will be collected through a list of observations: presence of depressive symptoms, socioeconomic status, education level, presence of comorbidity, polypharmacy, number of falls in the last year, number of hospitalizations in the last year and smoking habits. All data will be registered by a single researcher in a spreadsheet and then cross-checked point by point from the original data register by another researcher. Detailed information about general data collection can be found in Table A1.

The procedure for data collection will be performed in the following order: (i) clinical interview; (ii) index test; (iii) frailty assessment; (iv) sarcopenia assessment.

### 2.5. Index Test: Smartphone Mobile Application (App)

Participants’ sit-to-stand capacity will be assessed using an iPhone App called *Sit-to-Stand* (version 1.1.1.) installed on an iPhone 13 device running iOS 15.3; Apple Inc., Cupertino, CA, USA. This App was created to determine muscle power derived from one time rise from a chair using a slow-motion video captured camera (240 frames per second). Participants will sit on a rigid and adjustable height chair with their arms crossed over their chest with their hip, knee and ankle joints visually positioned at 90 degrees. Then, subjects will be instructed to stand from the chair “as fast as possible”, while the execution of the test will be filmed simultaneously with the smartphone. The smartphone will be placed horizontally on a 0.7 m-high tripod placed 3 m from the right or left side of the participant. This App estimates muscle power relative to body weight (in watts per kilogram; W/kg) from the subsequent analysis of the video recording using the following equation:*Muscle Power* = 2.773 − 6.228 × *t* + 18.224 × *d*
where *t* is the time to rising, i.e., the time from seat off until the hip and knee joints reach full extension in an upright stance, and *d* is the femur length of the subject, i.e., the vertical displacement during the sit-to-stand test. Femur length will be measured as the distance between the superior aspect of the greater trochanter and femoral lateral condyle on the participants’ right or left side. A reflective marker will be placed on the greater trochanter in order to identify the start and end of the sit-to-stand movement. The start of the movement is determined when the pelvis began to move forward after anterior trunk tilt which is time-matched when the reflective marker crossed the first horizontal grid line on the screen of the App (Figure 1). The end of the movement is defined when full extension of hip and knee is achieved in an upright stance which is time-matched when the reflective marker achieves the highest vertical point during the upright movement cycle. The results provided from this App has been twice validated against 3D motion capture camera and force plates in community-dwelling older adults [14,15].

In addition, participants’ calf circumference and mid-upper arm circumference will be measured and added together with sit-to-stand power values to improve diagnostic test accuracy. Calf circumference will be measured at the point of greatest circumference on the non-dominant leg (right leg for left-handed persons) in a sitting position with the knee and ankle at 90 degrees. Calf circumference has been shown to predict performance and survival in older people [17]. Mid-upper arm circumference will be measured at the midpoint of the participants’ upper arm between the tip of the acromion and olecranon process of the dominant arm in a sitting position [19]. Both anthropometric measurements will be measured in a relaxed state using an inelastic but flexible measuring tape without compressing the skin.

### 2.6. Reference Standard for Frailty Syndrome

Frailty syndrome will be assessed through frailty phenotype proposed by Linda Fried [20]. Although there is no gold standard for the detection of frailty, this tool is probably the most used in the scientific literature [3]. This multidimensional tool determines frailty through five domains: gait speed, exhaustion, unintentional weight loss, strength and physical activity levels. Gait speed will be assessed using the 4 m gait test at usual and comfortable pace. Unintentional weight loss will be assessed by asking the subject if he/she experienced unintentional weight loss over 4.5 kg through the previous year. Exhaustion assessment will be collected asking to participants to rate from 0 (never) to 3 (most of the time) the frequency with which they experience the following statements: “I felt that everything I did was an effort” and “I could not get going”. Strength level will be assessed using handgrip strength with a Takei 5401 hand grip digital dynamometer (Takei Scientific Instruments Co., Ltd., Tokyo, Japan) and data relative to the physical activity levels of the participants will be collected through the Spanish validated short version of the Minnesota leisure time physical activity questionnaire [21]. The presence of at least 3 of 5 criteria will be necessary to define a frail state, whereas the presence of at least 1 criterion will be sufficient to identify prefrailty state [20].

### 2.7. Reference Standard for Sarcopenia

The presence and the stage of sarcopenia will be determined following the recommendations provided by the European Working Group on Sarcopenia in Older People 2 (EWGSOP2) [2]. Accordingly, muscle strength will be determined through repeating sit-to-stand test five times, with a cut-off point higher than 15 s representing probable sarcopenia. Then, sarcopenia will be confirmed by appendicular skeletal muscle mass with a cut-off point lower than 20 and 15 kg for men and women, respectively. Appendicular skeletal muscle mass will be determined using bioelectrical impedance analysis (TANITA MC-580, Tanita Corp., Tokyo, Japan). Resistance index and reactance values will be used in the validated equation developed by Sergi et al. [22] as recommended by the EWGSOP2 [2]. Low physical performance will be assessed using the Short Physical Performance Battery test (SPPB) [23]. A score lower than 9 points will be indicative of low function and, therefore, severe sarcopenia. The EWGSOP2 recommends this tool for assessing the degree of severity of sarcopenia [2].

### 2.8. Sample Size Estimation

Since the prevalence of both syndromes (prefrailty and sarcopenia) in combination has been reported in a range from 13% to 40% in Spanish community-dwelling older adults [9,10], sample size was estimated according to the lowest prevalence rate to be more confident. Therefore, a minimum sample size of 200 participants per sex was estimated in accordance with Bujang and Adnan [24] to detect an effect size equal or higher than 0.80 area under the curve (AUC), with an alpha error of 0.05 and statistic power of 80%. Since the study aim is to develop a specific tool to be used as a diagnostic tool, a high degree of both sensitivity and specificity is usually necessary (for a detailed information, see Tables/Figures 1–3 in [24]).

### 2.9. Statistical Analysis

The database will be created in Microsoft Excel©, Microsoft Corp., Redmond, WA, USA and the software IBM SPSS Statistics 26.0, SPSS Inc., Chicago, Ill, USA will be used for further analysis. A descriptive analysis of data will be carried out to report the participants’ characteristics. We will provide values represented as mean, standard deviation, and 95% confidence interval (CI) for continuous variables and frequencies and percentages for categorical variables. Kolmogorov–Smirnov’s test will be used to assess the normal data distribution. A data scrub will be performed with range and logic tests and we will search for unknown or unlikely values.

A complete case analysis will be performed. The AUC of the receiver operating characteristic (ROC) will be calculated to identify the best cut-off point of muscle power to discriminate subjects with both syndromes separately and in combination. Then, a multiple logistic regression equation with muscle power and calf circumference or mid upper-limb circumference will be generated and an additional ROC analysis will be performed using predictive probability from the results of the multiple logistic regression analysis to identify the best cut-off point for detecting both syndromes for each sex. Cross-tabulations of the index test with the reference standards (prefrailty and sarcopenia syndromes) will be performed and sensibility, specificity, positive and negative predictive values, and positive and negative likelihood ratios will be calculated. Finally, after classifying the status of the subjects into robust, prefrail, sarcopenia or the combination of both syndromes, the chi-square tests will be used to assess differences in proportion regarding their clinical profile (Table A1) and the Odds Ratio with 95% CI will be calculated.

## 3. Discussion

For years, in order to prevent adverse health-related consequences in older adults, several studies have paid special attention to sarcopenia and frailty syndrome separately. Both syndromes have been used to categorize people according to their risk for developing disability, falls, hospitalizations, and mortality as well as to individualize appropriate interventions in order to revert these pathological conditions. Moreover, the phenotypic and criteria–diagnosis similarities between both syndromes have led to an undistinguished use of both concepts as if they were connected to one another. However, despite of the correlation between frailty and sarcopenia syndromes, recent studies have evidenced that they are distinct disorders [7,8,9,10]. In fact, whereas frailty individuals have shown two-fold increased mortality risk, frailty-sarcopenic individuals have shown more than three-fold mortality risk compared to neither frail nor sarcopenic individuals, highlighting that both syndromes act independently having an additive effect on mortality [7]. Furthermore, mortality risk has been recently reported to be superior in prefrail sarcopenic than prefrail non-sarcopenic individuals and prefrail individuals only experienced disability more frequently than robust one when they were also sarcopenic [10].

Given the growing interest for analyzing both syndromes in combination and the importance of the early identification of frailty (prefrailty status) and sarcopenia to prevent subsequent functional decline, disability, and the increased mortality risk, a clinical affordable easy-to-use diagnostic tool for detecting both syndromes in combination might have implications for prognosis, optimizing care, and planning interventions for community-dwelling older adults. Thus, the aim of this study is to develop and validate an update of the *Sit-to-Stand* App for detecting prefrailty and sarcopenia syndromes in combination in community-dwelling older adults.

Previous studies have analyzed the role of kinematic parameters during sit-to-stand test derived from sophisticated assessment tools such as force plates or inertial measurement units for detecting frailty and sarcopenia syndrome separately [12,13]. These studies have shown high specificity and sensitivity for detecting frailty or sarcopenia [12,13]; however, this technology, although sensitive to these syndromes, is not readily available or affordable for clinical or field-based testing environments and some current methods remain complex, difficult and time consuming to analyze. These characteristics represent barriers for clinicians that requires valid easy-to-use tools with automatic data processing and real-time results without complicated interpretation or instrumentation.

Recent advances in hand-held technology offer an opportunity to assess muscle power during sit-to-stand test through a smartphone mobile application (*Sit-to-Stand* App) based on high-speed video recording at 240 frames per second which can accurately detect the subject’s sit-to-stand motion [14,15]. This affordable technology only requires the use of a smartphone and an adjustable height chair to set starting knee joint position at 90 degrees for standardization purposes, since it is well known that chair height influences test performance [25,26,27]. This application has the potential to identify the rising phase of the sit-to-stand test, i.e., when the body loses contact with the seat and the center of mass is lifted until knee and hip extension is reached which is critical for muscle power calculation [28,29,30], otherwise muscle power could be underestimated. The application converts the time of this phase into muscle power relative to body weight through a regression equation which has been twice prospectively validated against 3D motion capture camera and force plates in community-dwelling older adults [14,15].

Since muscle mass has been related to frailty and sarcopenia syndrome and can be easily measured through calf circumference or upper mid-arm circumference [16,17], the integration of both outcomes (muscle mass and muscle power) into the mobile application will have the potential for detecting prefrailty and sarcopenia with enough accuracy. Thus, the update of this application will provide to the clinicians the ability to determine both syndromes either in combination or separately in a short time period with an easy-to-use interface and automatic data processing overcoming the aforementioned barriers.

Despite a rigorous approach towards data collection and synthesis, this research is not without limitations. First, the retrospective validation design of this study will limit the external validity of the mobile application for diagnosing purposes. However, prior to performing a prospective validation, a retrospective design is necessary. Second, since this study will be focused on people attending elderly social centers, these people are likely to be fitter or healthier than community-dwelling older adults who do not attend these centers. Therefore, the cut-off points estimated for diagnosing these syndromes in this research could be superior than those established in less fit people, so sensitivity and specificity might be influenced.

## 4. Conclusions

Considering the high socioeconomical impact of these syndromes, the creation of a highly reliable tool for detecting sarcopenia and prefrailty status will provide to the scientific community and clinical personnel the possibility to diagnose both syndromes with an easy-to-use interface and automatic data processing in short time periods within the environmental context of daily living. This innovative portable tool will have potential implications for prognosis, optimizing care, and planning interventions. In fact, it is well known that early detection of these syndromes might potentially increase the chances of reversibility.

## Figures and Tables

**Figure 1 sensors-22-06010-f001:**
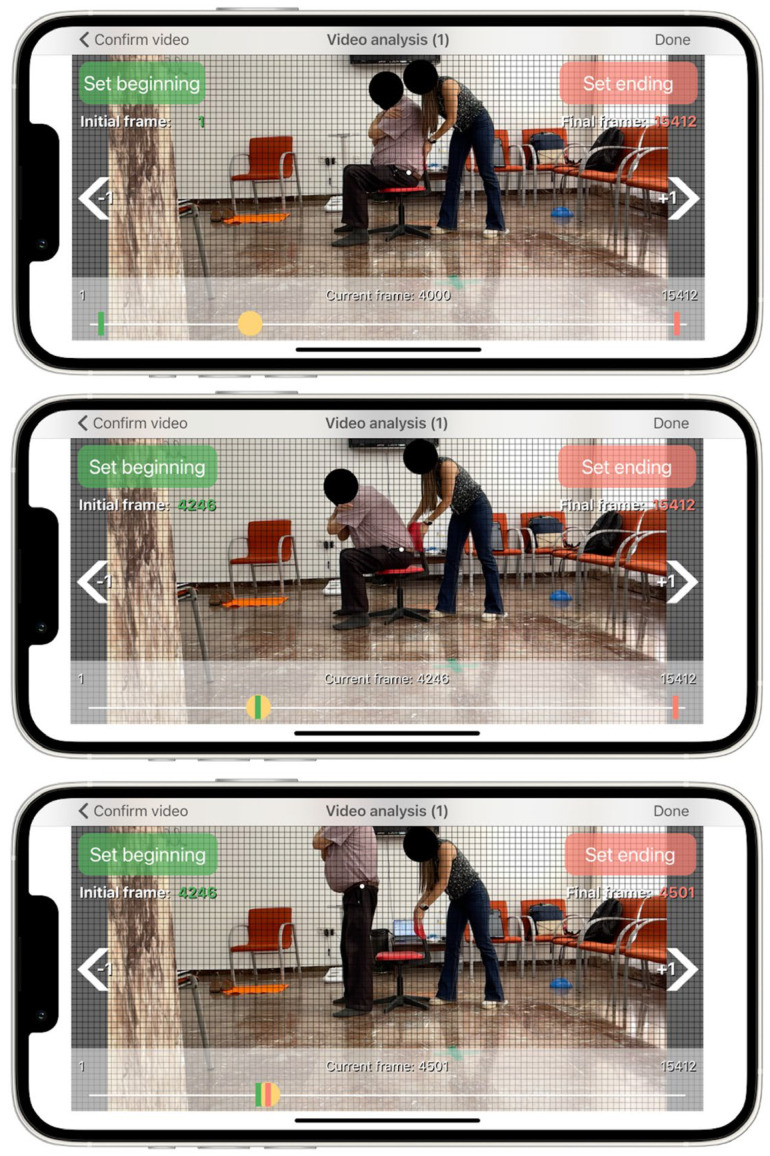
User interface of the App. White dot represents the reflective marker placed on the greater trochanter while the subject is in a sitting position (**top panel**), at the beginning of the vertical movement when the reflective marker crosses the first horizontal grip line on the screen (**middle panel**), and at the end of the vertical movement, when the reflective marker achieves the highest point (**lower panel**).

## Data Availability

Data will be made available upon request from the authors.

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
