# Peer review of "Validity of an iPhone App to Detect Prefrailty and Sarcopenia Syndromes in Community-Dwelling Older Adults: The Protocol for a Diagnostic Accuracy Study"

_sensors, 2022, doi:10.3390/s22166010_

Round 1

Reviewer 1 Report

GENERAL COMMENTS

The protocol is well written and the study will likely yield important data. Further methodological details are required to aid transparency and replication.

METHODS

Line 97: I cannot find the study on ClinicalTrials.gov using the ID NCT0514835. Please could you provide a link?

Eligibility criteria

Will you include people who require assistance to stand up from a chair?

Please clarify how the eligibility criteria will be assessed – presumably it will all be assessed via self-report?

Recruitment

How will you contact the potential participants? Email/telephone etc?

Line 116: Will data be collected on different days? If so, please provide more information on which measures will be taken on which days and in which order etc.

Line 130: How will you ensure participants’ hip, and ankle joints are at approximately 90 degrees? If you use the same height chair, this will vary considerably between individuals with different heights.

Lines 139-139: Please provide more information on how you will measure femur length i.e. what body landmarks will you use? Can you also provide more information on what the necessary procedures are to derive a measurement of muscle power from the app i.e. does the researcher need to self-identify the start and end of the sit-to-stand movement in the app?

Will you also measure sit-to-stand velocity as a secondary outcome?

Lines 141-146. Please provide more information on how mid-upper arm and calf circumference will be assessed. For example, the mid-upper arm measurement is often taken with a non-stretching measurement tape at the midpoint between the tip of the acromion and olecranon process of the dominant arm (e.g. see https://pilotfeasibilitystudies.biomedcentral.com/articles/10.1186/s40814-022-01069-1).

Line 150: “…the most used in the scientific literature”. Do you have a citation to support this?

Line 163: Do you have a citation to support using 3/5 criteria to indicate the presence of frailty?

Lines 171-172: Has the bioelectrical impedance equipment (TANITA MC-580) been validated?

Line 189: It isn’t the norm to report 95% confidence intervals for descriptive data. Is there a specific reason you plan to do this?

Line 202: Is “robust” the appropriate terminology? How about non-frail?

Please discuss your plans to make the raw data and SPSS syntax publicly available. This will be very important given the value of the dataset.

DISCUSSION

Line 254: Do you have a reference to support the statement that chair height influences performance?

Reviewer 2 Report

The crafting and sharing of this protocol will be of interest to gerontology researchers seeking to provide interventions for at-risk patients.  While the sensor component is quite subtle, the framework of analysis can be a useful tool for other researchers seeking to perform similar studies.  It would help those who are new to the field to have a slightly more detailed description of the sample size for statistical power, with the discussion of other methods presented.

Reviewer 3 Report

This paper seems incomplete, and I didn't find the experiment section. There seems to be content missing between pages 4 and 5. For the current version:

1. The introduction should emphasize existing problems along with existing literature and the purpose of the present study.

2. The whole framework is not adequately described, and the procedure used in this paper is by no means obvious. I suggest including a figure to describe it in detail.

3. Because I didn't find the important part of this paper, I cannot propose more suggestions or comments.

Round 2

Reviewer 3 Report

The manuscript is for a diagnostic accuracy study protocol, and no results are still available. I’m not sure whether it’s appropriate as a research article, but several problems should be addressed: 

1. The significance of the results should be discussed. Even though there are no quantitive results, you should be defining scientific truth.

2. As a research article, I think you should discuss the theoretical implications of your work.

3. Generally, the method must be reproducible, and statistical analyses are often necessary. I suggest you consider this issue. 
